**Data Availability Statement:** All relevant data are within the paper and its Supporting information files.

# Prevalence of *Toxoplasma gondii* in Galapagos birds: Inference of risk factors associated with diet

Juan D. Mosquera[1,2,3]*, Carlos A. Valle[3], Ainoa Nieto-Claudin[4,5], Birgit Fessl[4], Gregory A. Lewbart[3,6], Diane Deresienski[3,6], Leïla Bouazzi[7], Sonia Zapata[2,3], Isabelle Villena[1,8], Marie-Lazarine Poulle[1,9]

1 Epidémio-Surveillance et Circulation des Parasites dans les Environnements (ESCAPE), EA 7510, CAP SANTE, Université de Reims Champagne-Ardenne, Reims, France, 2 Instituto de Microbiología, Colegio de Ciencias Biológicas y Ambientales (COCIBA), Universidad San Francisco de Quito (USFQ), Quito, Ecuador, 3 Galápagos Science Center (GSC), Colegio de Ciencias Biológicas y Ambientales (COCIBA), Universidad San Francisco de Quito (USFQ), Quito, Ecuador, 4 Charles Darwin Research Station, Charles Darwin Foundation, Santa Cruz, Galapagos Islands, Ecuador, 5 Saint Louis Zoo Institute for Conservation Medicine, Saint Lois Zoo, Saint Louis, Missouri, United States of America, 6 College of Veterinary Medicine, North Carolina State University, Raleigh, North Carolina, United States of America, 7 Comité Universitaire de Ressources pour la Recherche en Santé, Université de Reims Champagne-Ardenne, Reims, France, 8 Laboratoire de Parasitologie-Mycologie, Centre National de Référence de la Toxoplasmose, Centre de Ressources Biologiques *Toxoplasma*, CHU Reims, Reims, France, 9 CERFE, Université de Reims Champagne-Ardenne, Boult-aux-Bois, France

* jdmosquera@usfq.edu.ec

## Abstract

*Toxoplasma gondii* is a zoonotic intracellular parasite of particular concern in the conservation of wildlife due to its ability to infect all homeotherms and potentially cause acute fatal disease in naive species. In the Galapagos (Ecuador), an archipelago composed of more than a hundred islets and islands, the presence of *T. gondii* can be attributed to human-introduced domestic cats, but little is known about its transmission in wildlife populations. We compared the prevalence of antibodies against *T. gondii* in sympatric Galapagos wild bird species that differ in diet and contact with oocyst-contaminated soil to determine the relative importance of trophic habits as an exposure factor. Plasma samples were obtained from 163 land birds inhabiting Santa Cruz, one of the cat-inhabited islands, and from 187 seabirds breeding in cat-free surrounding islands (Daphne Major, North Seymour, and South Plaza). These samples were tested for the presence of *T. gondii* antibodies using the modified agglutination test (MAT ≥ 1:10). All seven species of land birds and 4/6 species of seabirds presented seropositive results. All great frigatebirds (*Fregata minor*) (N = 25) and swallow-tailed gulls (*Creagrus furcatus*) (N = 23) were seronegative. Prevalence ranged from 13% in Nazca boobies (*Sula granti*) to 100% in Galapagos mockingbirds (*Mimus parvulus*). It decreased from occasional carnivores (63.43%) to granivores-insectivores (26.22%), and strict piscivores (14.62%). These results indicate that the consumption of tissue cysts poses the highest risk of exposure to *T. gondii* for Galapagos birds, followed by the ingestion of plants and insects contaminated by oocysts as important transmission pathways.

**Funding:** JM and SZ - The Galápagos Science Center-USFQ/University of North Carolina-Chapel Hill https://www.galapagosscience.org/ JM, IV and MLP - Université de Reims Champagne Ardenne https://www.univ-reims.fr/ NO The funders had no role in study design, data collection and analysis, decision to publish, or preparation of the manuscript.

**Competing interests:** The authors have declared that no competing interests exist.

## Introduction

The trophic habits of wildlife species can be a predictor of exposure to pathogens shared with humans and domestic animals [1–3]. Therefore, characterizing the relationship between diet and parasite transmission is an asset in understanding and preventing parasitic infections that affect wildlife populations of conservation concern. This is particularly relevant for toxoplasmosis, a zoonotic infection transmitted in a prey-predator cycle that can affect any homeotherm [4, 5]. Toxoplasmosis can be a serious threat to wildlife species that have not coevolved with this parasite [6, 7].

The causative agent of toxoplasmosis is *Toxoplasma gondii*, a protozoan of the Coccidia subclass. While most coccidia are monoxenous, *T. gondii* alternate sexual and asexual reproduction in different hosts species [8, 9]. The sexual cycle is carried exclusively in felids (the definitive host taxon) and results in the excretion of oocysts, the environmentally resistant infective forms [9]. Intermediate hosts (birds and mammals) that ingest such oocysts develop tissue cysts containing bradyzoites (asexual stages) in muscles and other organs [9]. The consumption of tissue cysts by a definitive host completes the cycle [9]. *Toxoplasma gondii* can also propagate asexually between intermediate hosts that consume tissue cysts from infected animals. Furthermore, many heterotherms can carry and transmit *T. gondii* oocysts, serving as potential paratenic hosts, as demonstrated experimentally for dung beetles [10], flies [11–13], cockroaches [14, 15], earthworms [16] or fish [17].

Even when infection by *T. gondii* can be asymptomatic in most avian species, birds are of great epidemiological importance among intermediate hosts, since they act as reservoirs and dispersers [18, 19]. Moreover, ground-foraging birds are used as sentinels of oocysts-contaminated soils, while prevalence in raptors signals the infection of rodents and other small mammals [20, 21]. Exposure of wild birds to *T. gondii* is common in certain trophic groups [19, 22, 23]. Remarkable high seroprevalence has been reported in raptors and scavengers with varied diets from America and Europe [24, 25]. However, apart from the studies conducted by Cabezón et al. [26] in Spain and Chen et al. [27] in Taiwan, little information is available on the prevalence of *T. gondii* in different trophic groups of birds from the same geographical area. Furthermore, the results from Spain and Taiwan were not consistent: in Spain, the highest prevalence was found in raptors (36.2%), while in Taiwan the prevalence was highest in piscivores and omnivores (34.29%). More information on seroprevalence in birds with different diets and from the same local geographic area is needed to better comprehend the trophic component of exposure risk to *T. gondii* in wild birds. The high diversity of endemic bird populations in the Galapagos and the geographical isolation of this archipelago in the Pacific Ocean offer the opportunity to collect this type of data. Galapagos bird populations have declined in numbers and range over the past 20 years, and threats such as habitat degradation, introduced diseases, and invasive species are among the main factors involved in this decline [28, 29]. Galapagos was feline-free before the introduction of the domestic cat (*Felis silvestris catus*) in the 19th century [30]. Since then, domestic cats are the only definitive host of *T. gondii* in the archipelago and after the eradication from Baltra Island [31] their presence is limited to the four human-inhabited islands. The recent and spatially restricted introduction of cats raises concern about the susceptibility of Galapagos wild birds to *T. gondii* and the trophic habits associated with exposure risk.

This study aims to estimate the prevalence of antibodies against *T. gondii* in Galapagos wild birds to infer the relative importance of oocyst-contaminated soil and hosts-type (intermediate or paratenic hosts) in their diets as transmission routes of *T. gondii*. Our results contribute to the formation of a health baseline for endemic birds that can be further used for other endangered species.

## Materials and methods

### Ethics statement

Bird handling and blood sampling were performed by highly trained professionals and followed standard vertebrate protocols and veterinary practices. All procedures were conducted in accordance with the regulations of the Galapagos National Park Directorate and under their approval (PC 01–18, PC 30–19, PC 06–21, PC 59–17 and PC 04–22). This publication is contribution number 2500 of the Charles Darwin Foundation for the Galapagos Islands.

### Study sites

The Galapagos archipelago comprises more than 100 islands and islets of volcanic origin situated in the Eastern Tropical Pacific Ocean, at approximately 1000 km from continental Ecuador. Climate is subtropical with a hot season from January to May and a cool season from June to December [32]. The barren landscape dominates, except at higher altitudes that receive enough rain to support a lush, tropical environment. Sampling was conducted in the most human and cat-populated island (Santa Cruz, 986 km$^2$) and three surrounding islands free of human and cat populations: Daphne Major (0.33 km$^2$), North Seymour (1.9 km$^2$) and South Plaza (0.12 km$^2$) (Fig 1).

### Sample collection

Seabirds and land birds' samples were initially collected for other studies between 2017 and 2022.

**Land birds.** Seven endemic bird species of the order Passeriformes were sampled in Santa Cruz for a total of 163 individuals (Table 1). Birds were caught with mist nets. Blood samples ($< 1\%$ of body weight) were collected from the ulnar vein using a 25 or 26-gauge needle by pricking the vein and then filling 1–2 heparinized capillary tubes. The birds were briefly held until the bleeding stopped. Handling took less than 15 minutes and birds were safely released thereafter. Plasma and blood cells were separated by centrifugation at the Charles Darwin Research Station (CDRS) laboratory on the same day of collection. Samples were stored in 1.5 ml cryotubes at -20˚C until further processing. Given the small amounts of plasma obtained, most samples from the same species caught the same day were grouped in pools (Table 1).

**Seabirds.** Six species of seabirds were sampled in Daphne Major, North Seymour, and South Plaza islands for a total of 187 individuals (Table 2). As seabirds are relatively sedentary on land, they were captured from their nesting holes or nests, one at a time. They were released at their capture site just after a blood sample was obtained quickly and safely within 10 minutes. Brachial venipuncture was performed using a heparinized 25-gauge needle and 1 ml syringe to collect up to 1 ml of blood per bird. Blood samples were transported to the Galapagos Science Center at San Cristobal Island in a cooler with ice packs. Plasma was obtained by centrifugation and stored at -20˚C until serological analyses.

**Feeding type determination.** Six main feeding types were determined based on the literature for the 13 bird species studied (Table 3). Species were then grouped into three classes according to the source of exposure to *T. gondii* associated with their diet. Galapagos mockingbirds (*Mimus parvulus*) and magnificent frigatebirds (*Fregata magnificens*) were grouped as occasional carnivores since they are exposed to infected tissues and oocysts in both terrestrial and marine environments. Yellow warblers (*Setophaga petechia aureola*), Galapagos flycatchers (*Myiarchus magnirostris*), green warbler finches (*Certhidea olivacea*), medium ground

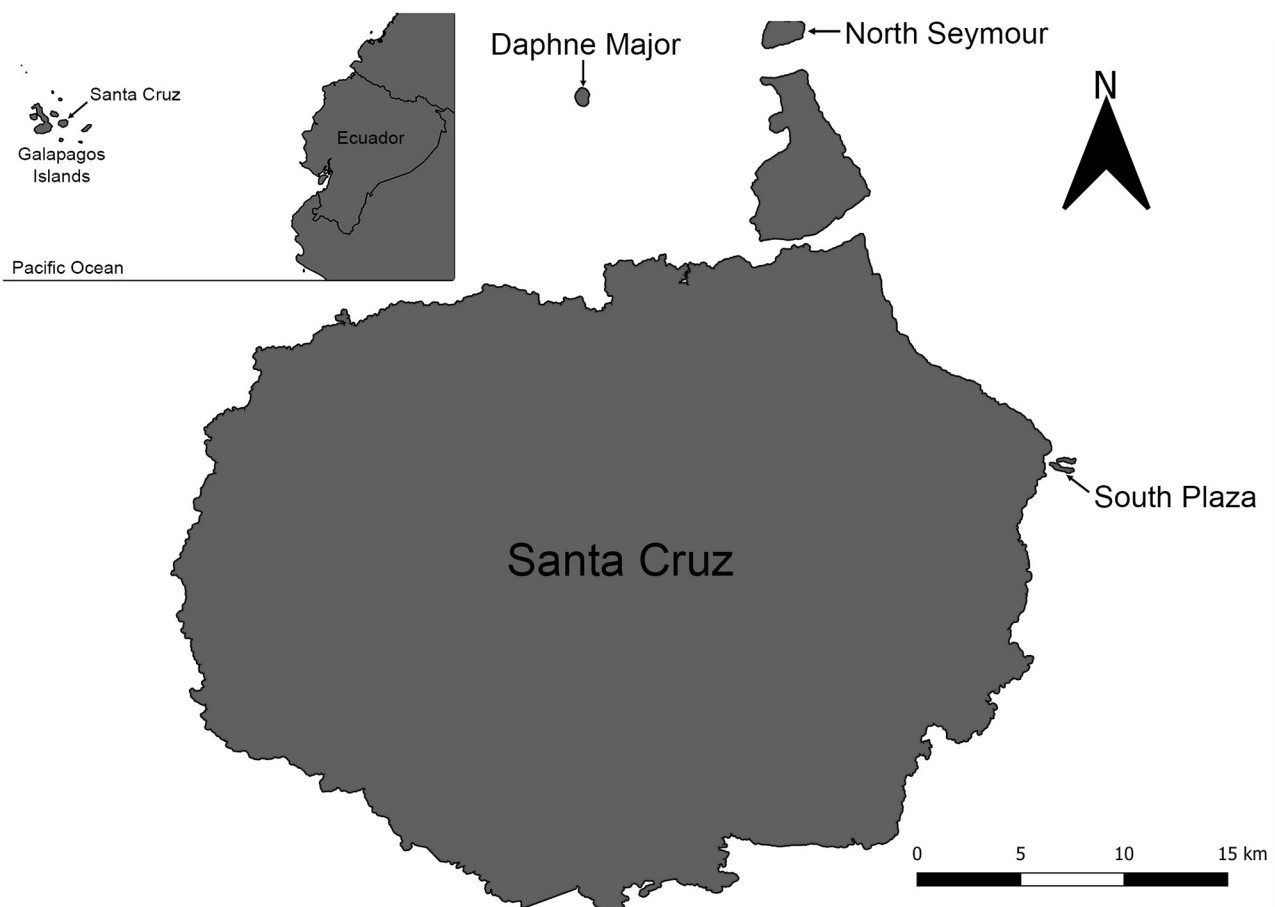

**Fig 1. Location of the Galapagos archipelago and the four Galapagos islands where birds were sampled for the detection of *T. gondii* antibodies.**
Land birds were sampled on cat-inhabited Santa Cruz, and seabirds on the surrounding cat-free islands: Daphne Major, North Seymour and South Plaza. Map created using QGIS 3.22.7 and modified from base map OSM Standard (OpenStreetMap openstreetmap.org/copyright).

**Table 1. Land bird sampling in Santa Cruz Island (Galapagos archipelago) and prevalence of antibodies against *T. gondii* per species as determined using the modified agglutination test (MAT).**

| Species | No. tested | Pool sizes (Positive in bold MAT ≥ 1:10) | Prevalence % [CI 95%] MAT ≥ 1:10 | Prevalence % [CI 95%] MAT ≥ 1:40 |
|---|---|---|---|---|
| Galapagos mockingbird (*Mimus parvulus*) | 13 | **3, 3, 2, 2, 2, 1** | 100.00 | 100.00 |
| Green warbler finch (*Certhidea olivacea*) | 5 | **3, 2** | 100.00 | 22.54 [1.43–68.83] |
| Yellow warbler (*Setophaga petechia aureola*) | 22 | **4, 3, 3**, 3, **2, 2, 2, 2**, 1 | 49.09 [23.63–77.33] | 35.27 [15.49–60.29] |
| Small ground finch (*Geospiza fuliginosa*) | 36 | **6, 5**, 5, **4**, 3, 3, **2, 2, 2**, 2, **1**, 1 | 30.64 [14.95–51.12] | 16.57 [6.23–32.75] |
| Galapagos flycatcher (*Myiarchus magnirostris*) | 32 | 5, **4**, 4, **3, 3**, 3, 3, 2, 2, **1** | 22.99 [9.73–41.83] | 0.00 |
| Small tree finch (*Camarhynchus parvulus*) | 21 | 3, **3, 2, 2**, 2, 2, 2, 2, 2, 1 | 21.37 [7.14–43.24] | 10.03 [1.74–27.92] |
| Medium ground finch (*Geospiza fortis*) | 34 | **4, 4**, 4, 4, 4, **3**, 3, 3, **2**, 2, 1 | 13.80 [4.47–29.5] | 6.27 [1.07–18.17] |
| **Overall** | **163** | | **29.31 [21.60–38.00]** | **15.16 [9.93–21.64]** |

**Table 2. Seabird sampling in South Plaza, North Seymour and Daphne Major Islands (Galapagos archipelago) and prevalence of antibodies against *T. gondii* per species as determined using the modified agglutination test (MAT).**

| Species | No. tested | No. positives MAT ≥ 1:10 | Prevalence % [CI 95%] MAT ≥ 1:10 | Prevalence % [CI 95%] MAT ≥ 1:40 |
|---|---|---|---|---|
| Magnificent frigatebird (*Fregata magnificens*) | 16 | 9 | 56.25 [33.18–76.9] | 37.50 [18.48–61.36] |
| Blue-footed booby (*Sula nebouxii*) | 45 | 14 | 31.11 [19.5–45.66] | 6.67 [2.29–17.86] |
| Red-billed tropicbird (*Phaeton aethereus*) | 40 | 6 | 15.00 [7.06–29.07] | 5.00 [1.38–16.50] |
| Nazca booby (*Sula granti*) | 38 | 5 | 13.16 [5.75–27.33] | 7.89 [2.72–20.08] |
| Swallow-tailed gull (*Creagrus furcatus*) | 23 | 0 | 0.00 [0.00–14.31] | 0.00 [0.00–14.31] |
| Great frigatebird (*Fregata minor*) | 25 | 0 | 0.00 [0.00–13.32] | 0.00 [0.00–13.32] |
| **Overall** | **187** | **34** | **18.20 [13.30–24.30]** | **7.49 [4.51–12.17]** |

finches (*Geospiza fortis*), small ground finches (*Geospiza fuliginosa*), and small tree finches (*Camarhynchus parvulus*) were grouped as granivores-insectivores. They are exposed to oocysts in terrestrial environments. Blue-footed boobies (*Sula nebouxii*), red-billed tropicbirds (*Phaeton aethereus*), Nazca boobies (*Sula granti*), swallow-tailed gulls (*Creagrus furcatus*), and great frigatebirds (*Fregata minor*) were grouped as strict piscivores. They are only exposed to oocysts in marine environments.

## Serological analyses

Plasma samples were examined using the Modified Agglutination Test (MAT) for the detection of antibodies against *T. gondii* [48]. This test is the most sensitive and specific for the detection of antibodies against *T. gondii* in birds [22, 49]. Formalized tachyzoites produced at the Laboratory of Parasitology, National Centre on Toxoplasmosis (Reims, France) were used as antigen. All samples were tested at 1:10, 1:20, 1:40, 1:80, 1:160 and 1:320 dilutions. Agglutination at 1:10 or higher was considered as positive. Prevalence at MAT ≥ 1:40 was also calculated for all bird species to allow comparisons between previous studies conducted on Galapagos birds based on MAT ≥ 1:50 [50, 51].

**Table 3. Main feeding types of sampled Galapagos bird species as determined from literature.**

| Species | Main feeding type(s) | Meat | Insects | Seeds & fruits | Fish | References |
|---|---|---|---|---|---|---|
| Galapagos mockingbird *Mimus parvulus* | Occasional carnivore (Omnivore) | + | +++ | ++ | | Grant et al [33] Curry et al [34] Gotanda et al [35] |
| Magnificent frigatebird *Fregata magnificens* | Occasional carnivore (Mainly piscivore) | + | | | +++ | Calixto-Albarrán et al [36] |
| Green warbler finch *Certhidea olivacea* | Insectivore | | +++ | + | | Filek et al [37] |
| Yellow warbler *Setophaga petechia aureola* | Insectivore | | +++ | + | | Guerrero et al [38] |
| Galapagos flycatcher *Myiarchus magnirostris* | Insectivore | | +++ | + | | Tebbich et al [39] |
| Medium ground finch *Geospiza fortis* | Granivore | | + | +++ | | De León et al [40] |
| Small ground finch *Geospiza fuliginosa* | Granivore | | + | +++ | | Boag et al [41] |
| Small tree finch *Camarhynchus parvulus* | Insectivore & granivore | | +++ | +++ | | Tebbich et al [39] |
| Red-billed tropicbird *Phaeton aethereus* | Strict piscivore | | | | +++ | Castillo-Guerrero et al [42] |
| Blue-footed booby *Sula nebouxii* | Strict piscivore | | | | +++ | Anchundia et al [43] |
| Nazca booby *Sula granti* | Strict piscivore | | | | +++ | García et al [44] |
| Swallow-tailed gull *Creagrus furcatus* | Strict piscivore | | | | +++ | Harris et al [45] Hailman et al [46] |
| Great frigatebird *Fregata minor* | Strict piscivore | | | | +++ | Harrison et al [47] |

## Statistical analyses

For individual data collected on seabirds, the prevalence of *T. gondii* antibodies per species was calculated conventionally as n/N, where n represents the number of positive individuals and N the total number of birds sampled in the species. For pooled data collected on land birds, the prevalence of *T. gondii* antibodies per species was calculated using the method of William and Moffit based on maximum likelihood and assuming perfect sensitivity and specificity of the serological test [52]. A log-linear Poisson regression was used to compare the results according to the three feeding type classes (occasional carnivores/granivores-insectivores/strict piscivores). The total number of bird species for each feeding category was converted to a logarithm and included as an offset in the model. SAS software version 9.4 (SAS Institute, Cary, NC) was used to perform data analysis, with 0.05 as significance level for statistical tests.

## Results

Seropositivity was found in all studied land bird species (Table 1) with an overall prevalence of 29.31% (CI$_{95\%}$: 21.60–38.00). The highest prevalence was found in Galapagos mockingbirds and green warbler finches (100%), and the lowest in medium ground finches (13.80%, CI$_{95\%}$: 4.47–29.5) (Table 1). Seropositivity was found in four out of six seabird species with an overall prevalence in the sampled seabird population of 18.18% (CI95%: 13.30–24.30) (Table 2). Prevalence ranged from 56.25% (CI$_{95\%}$: 33.18–76.9) in magnificent frigatebirds to 0% in both great frigatebirds (CI$_{95\%}$: 0.00–13.32) and swallow-tailed gulls (CI$_{95\%}$: 0.00–14.31) (Table 2). Prevalence was lower in most seropositive land birds and seabirds when considering a MAT ≥ 1:40, with Galapagos flycatchers becoming seronegative (Tables 1 & 2). Prevalence varied significantly according to the three feeding types (Wald Chi-Square = 12,86; p = 0,002). It was highest in occasional carnivores (63.43%, CI$_{95\%}$: 42.55–81.8) and decreased to granivores & insectivores (26.22%, CI$_{95\%}$: 18.71–34.92) and the strict piscivores (14.62%, CI$_{95\%}$: 10.10–20.69) (Fig 2).

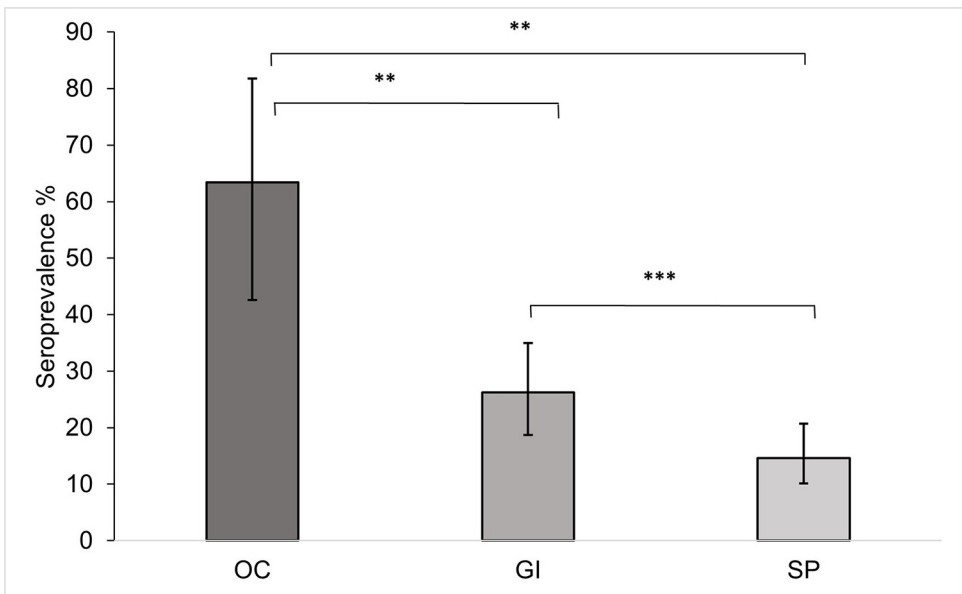

**Fig 2. Seroprevalence of antibodies against *T. gondii* by feeding type of Galapagos birds (95% CI).** OC: occasional carnivores, GI: granivores-insectivores, SP: strict piscivores (SP), ** and ***: difference at p < 0.05 and p < 0.005 respectively.

## Discussion

The present study reports the presence of antibodies against *T. gondii* in Galapagos land birds and seabirds with a significant variation in exposure according to their feeding type. The prevalence values in occasional carnivore birds were much higher than those previously reported for Galapagos hawks, *Buteo galapagoensis* (2% of 48 at MAT ≥ 1:50 in Deem et al. [51] *versus* 100% in Galapagos mockingbirds and 37.50% in magnificent frigatebirds at MAT ≥ 1:40 in this study). The Galapagos mockingbird and the magnificent frigatebird, land bird and seabird respectively but both with feeding habits that include the sporadic consumption of meat and other derivates from potential intermediate hosts, had a significantly higher prevalence than granivores-insectivores and strict piscivores. The Galapagos mockingbird is a predominately insectivorous and fruit-eating species, searching much of its food on the ground. They are known to feed occasionally on small vertebrates including mammals, birds and sea lion placenta (*Zalophus wollebaeki*) [33, 35]. Similarly, even if mainly piscivore, the magnificent frigatebird can feed on chicks and carrion that includes sea lion placenta [53, 54]. Such animal-derived dietary components, potential sources of infection by *T. gondii*, could explain the elevated seropositivity. Coinciding results have been reported in carnivore and omnivore mammals compared to herbivore and insectivore species, likely due to a higher probability of consuming infected tissues than ingesting oocysts from contaminated soil or water [55–57]. The lower prevalence found in Galapagos hawks compared to our sampled birds may be simply explained by sample sites: all hawks were sampled in the cat-free Santiago Island, where they might exclusively hunt while the Galapagos mockingbirds and magnificent frigatebirds in our study probably fed mainly on or close to Santa Cruz, a cat-inhabited island.

Seropositive results found in all species of granivore and insectivore wild birds sampled in Santa Cruz signal the presence of *T. gondii* oocysts in the soil, water, and insects of this island. By comparison, the prevalence found in the strict piscivore seabird species (14.60%) was significantly the lowest among all the categories but higher than the one previously reported in two other Galapagos piscivorous birds (2.3%): Galapagos penguin (*Spheniscus mendiculus*) and flightless cormorant (*Phalacrocorax harrisi*) [50]. As with Galapagos hawks, agglutination at 1:50 sera dilutions or higher was considered as positive for the MAT test in these two seabird species which could have led to an underestimation of prevalence. It is worth noting that, even when considering a similar dilution (MAT ≥ 1:40), the prevalence found in strict piscivore seabirds in this study was still higher than the one reported in Galapagos penguins and flightless cormorants from Isabela (cat-inhabited) and Fernandina (cat-free) islands. The influx of oocysts towards the ocean might be exacerbated by the abundant cat populations and anthropogenic activities in Santa Cruz due to being the most populated island. This could result in a greater exposure to *T. gondii* for seabirds from islands surrounding Santa Cruz than in seabirds from Isabela and Fernandina and would explain the differences in prevalence.

None of the sampled great frigatebirds and swallow-tailed gulls were seropositive for *T. gondii* antibodies. These two species are highly pelagic and feed mainly on flying fish and squids [45–47, 54] that, to our knowledge, are not paratenic hosts of *T. gondii*. It is interesting to note that, similarly, none of the great frigatebirds sampled in the Western Indian Ocean were exposed to *T. gondii* [58]. By contrast, we detected antibodies against *T. gondii* in Nazca boobies, blue-footed boobies, and red-billed tropicbirds. These seabirds feed on fish from the families Clupleidae (sardines and herrings) and Engraulidae (anchovies) [42–44] that have the ability to retain and transport *T. gondii* oocysts [17, 59]. However, the prevalence found in blue-footed boobies (31.11%) doubles the prevalence of Nazca boobies (15%) and red-billed tropic birds (13.16%). This difference might be due to differences in foraging zones. Coastal foragers like blue-footed boobies may be at a higher risk of exposition to ocean contamination

by oocysts carried from Santa Cruz via runoff [60] than pelagic birds like Nazca boobies and red-billed tropic birds. Similarly, coastal foragers were more exposed to *T. gondii* than pelagic ones in the seabirds from the Western Indian Ocean [58]. To our knowledge, this study is the first to report the presence of antibodies against *T. gondii* in seven endemic Galapagos land bird species and three seabird species.

It also supports that the consumption of tissue cysts is the main exposure route to *T. gondii* for Galapagos birds. The elevated prevalence in occasional carnivores suggests that *T. gondii* is already infecting other intermediate hosts in the archipelago. Ingestion of oocysts from contaminated food (vegetation and insects) in islands with cats is identified as the second main exposure factor, followed by the consumption of paratenic host fish for strict piscivore birds. As fatal toxoplasmosis has been reported in nene geese (*Nesochen sandvicensis*), red-footed boobies (*Sula sula*) and crows (*Corvus hawaiiensis*) from Hawaii where domestic cats were introduced at a similar time as in the Galapagos Islands [61, 62], our results raise questions about the risk of acute forms of disease in Galapagos birds. *Toxoplasma gondii* adds up to the list of potential threats of the already endangered Galapagos bird populations that can be aggravated by their insular naivety [29]. Our study helps highlight a potential new health risk and to further develop a health baseline that can also include endangered species to strengthen control policies regarding the anthropogenic introduction of alien species and their pathogens. Further research is needed to better understand the impact of *T. gondii* on the Galapagos fauna and its circulation between islands with and without cats.

## Supporting information

**S1 Table. Information on land birds and seabirds sampled in the Galapagos Islands for the detection of antibodies against *Toxoplasma gondii* in their sera.**
(XLSX)

## Acknowledgments

We thank David Anchundia and Courtney Pike (CDF) for the serum collection of land bird species. We also thank the staff of the Galápagos Academic Institute for the Arts and Sciences (GAIAS)-Universidad San Francisco de Quito (USFQ) and the Galápagos Science Center-USFQ/University of North Carolina-Chapel Hill for their support and assistance with sample handling.

## Author Contributions

**Conceptualization:** Juan D. Mosquera, Carlos A. Valle, Sonia Zapata, Isabelle Villena, Marie-Lazarine Poulle.

**Data curation:** Juan D. Mosquera.

**Formal analysis:** Juan D. Mosquera, Leïla Bouazzi.

**Funding acquisition:** Juan D. Mosquera, Sonia Zapata, Isabelle Villena, Marie-Lazarine Poulle.

**Investigation:** Juan D. Mosquera.

**Methodology:** Juan D. Mosquera, Leïla Bouazzi, Marie-Lazarine Poulle.

**Resources:** Birgit Fessl, Gregory A. Lewbart, Diane Deresienski.

**Supervision:** Sonia Zapata, Isabelle Villena, Marie-Lazarine Poulle.

**Writing – original draft:** Juan D. Mosquera.

**Writing – review & editing:** Juan D. Mosquera, Carlos A. Valle, Ainoa Nieto-Claudin, Birgit Fessl, Gregory A. Lewbart, Leïla Bouazzi, Sonia Zapata, Isabelle Villena, Marie-Lazarine Poulle.

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
