## [Decision Letter · Decision Letter 0]

3 May 2023

PONE-D-23-07690Prevalence of Toxoplasma gondii in Galapagos birds: inference of risk factors associated with dietPLOS ONE

Dear Dr. Mosquera,

Thank you for submitting your manuscript to PLOS ONE. After careful consideration, we feel that it has merit but does not fully meet PLOS ONE’s publication criteria as it currently stands. Therefore, we invite you to submit a revised version of the manuscript that addresses the points raised during the review process.

Please make the minor corrections suggested by the reviewers. If you do not agree with any suggestion by any of the reviewers, please provide an appropriate response to the comments. 

We look forward to receiving your revised manuscript.

Kind regards,

Christopher Adenyo, Ph.D.

Academic Editor

PLOS ONE

Journal Requirements:

"JM and SZ - The Galápagos Science Center-USFQ/University of North Carolina-Chapel Hill

https://www.galapagosscience.org/

JM, IV and MLP - Université de Reims Champagne Ardenne

https://www.univ-reims.fr/

NO"

"..This research was authorized by the Galápagos National Park Service and was conducted with support of the Galápagos Academic Institute for the Arts and Sciences (GAIAS)-Universidad San Francisco de Quito (USFQ) and the Galápagos Science Center-USFQ/University of North Carolina-Chapel Hill.."

"JM and SZ - The Galápagos Science Center-USFQ/University of North Carolina-Chapel Hill

https://www.galapagosscience.org/

JM, IV and MLP - Université de Reims Champagne Ardenne

https://www.univ-reims.fr/

NO"

6. We note that Figure 1 in your submission contain map images which may be copyrighted. All PLOS content is published under the Creative Commons Attribution License (CC BY 4.0), which means that the manuscript, images, and Supporting Information files will be freely available online, and any third party is permitted to access, download, copy, distribute, and use these materials in any way, even commercially, with proper attribution. For these reasons, we cannot publish previously copyrighted maps or satellite images created using proprietary data, such as Google software (Google Maps, Street View, and Earth). For more information, see our copyright guidelines: http://journals.plos.org/plosone/s/licenses-and-copyright.

(1) You may seek permission from the original copyright holder of Figure 1 to publish the content specifically under the CC BY 4.0 license.  

**Additional Editor Comments:**

I received reviews from two experts in the study of T. gondii. Both reviewers recommended that the manuscript should be accepted. I am therefore happy to inform you that your manuscript can be published after minor edits suggested by the reviewers. Please provide responses to the issues raised by reviewer #1.

Reviewers' comments:

Reviewer's Responses to Questions

**Comments to the Author**

1. Is the manuscript technically sound, and do the data support the conclusions?

Reviewer #1: Yes

Reviewer #2: Yes

2. Has the statistical analysis been performed appropriately and rigorously? 

Reviewer #1: I Don't Know

Reviewer #2: N/A

3. Have the authors made all data underlying the findings in their manuscript fully available?

Reviewer #1: Yes

Reviewer #2: Yes

4. Is the manuscript presented in an intelligible fashion and written in standard English?

Reviewer #1: Yes

Reviewer #2: Yes

5. Review Comments to the Author

Reviewer #1: The authors sought to find the prevalence of T. gondii in the Galapagos birds and how the feeding behaviour of the birds influences their exposure to T. gondii infection. They found that occassionally carnivorous birds had the highest exposure to the parasite, followed by the granivore-insectivores and lastly the piscivores.

The author's sampled an appreciable number of birds to make a good inference.

I am not an expert in statistics, but I could understand the authors' analysis of the data.

I have no major correction to make on this paper.

Minor Comments

1. LIne 117 & Line 139-140 Is there a significance to the different storage conditions for the land birds and sea birds i.e. -80 and -20 degrees celcius respectively

2. Line 137-138; the authors' mentioned recording the physical conditions, age and sex of the seabirds. However, this information was not used in any analysis.

3. Line 255: Authors' should kindly reconstruct the sentence to provide more clarity.

Reviewer #2: This is unique study of T. gondii prevalence in birds from 2 areas with or without cats. The serologic test used is the most specific tests for T. gondii infection in cats. The results ae of great scientific interest. The authors should consider replacing toxoplasmosis (clinical) with T. gondii infection.

6. PLOS authors have the option to publish the peer review history of their article (what does this mean?). If published, this will include your full peer review and any attached files.

Reviewer #1: No

Reviewer #2: **Yes: **Jitender P Dubey

---

## [Author Response · Author response to Decision Letter 0]

31 May 2023

Dear Editors and Reviewers,

Thank you very much for your comments and observations on the manuscript.

Here you will find the responses to each point raised:

Journal Requirements:

1. We have ensured that the manuscript meets PLOS ONE's style requirements.

2. We have provided additional information regarding the handling of the animals and the efforts to alleviate suffering.

3. We have included the statement “The funders had no role in study design, data collection and analysis, decision to publish, or preparation of the manuscript.” in the cover letter.

4. We have removed the funding-related text from the manuscript and would like to remain with the original Funding Statement since the people mentioned are part of The Galápagos Science Center-USFQ/University of North Carolina-Chapel Hill. 

5. We have uploaded the study’s minimal underlying data set as Supporting Information.

6. We remade Figure 1 using QGIS and the base map OSM Standard from OpenStreeMap. We have included the required text that provides credit to OpenStreetMap by displaying their copyright notice following their attribution guidelines.

7. We have reviewed the reference list to ensure that it is complete and correct.

Reviewers’ comments:

• We have addressed the reviewers’ minor comments and are marked as track changes in the revised version of the article. 

• We have corrected the storage conditions of the samples (all samples were stored at -20°C, -80C was written by mistake).

• The recording of the physical conditions, age and sex of the seabirds is no longer mentioned in the article since it was not used in our study.

• We have reconstructed the sentence in line 255.

• We have addressed the difference between Toxoplasmosis and T. gondii infection.

---

## [Editor Report · Decision Letter 1]

5 Jun 2023

Prevalence of Toxoplasma gondii in Galapagos birds: inference of risk factors associated with diet

PONE-D-23-07690R1

Dear Dr. Mosquera,

We’re pleased to inform you that your manuscript has been judged scientifically suitable for publication and will be formally accepted for publication once it meets all outstanding technical requirements.

Kind regards,

Christopher Adenyo, Ph.D.

Academic Editor

PLOS ONE
---

## [Editor Report · Acceptance letter]

14 Jun 2023

PONE-D-23-07690R1 

Prevalence of *Toxoplasma gondii* in Galapagos birds: inference of risk factors associated with diet 

Dear Dr. Mosquera:

I'm pleased to inform you that your manuscript has been deemed suitable for publication in PLOS ONE. Congratulations! Your manuscript is now with our production department. 

Kind regards, 

on behalf of

Dr. Christopher Adenyo 

Academic Editor

PLOS ONE